

# Dementia-related user-based collaborative filtering for imputing missing data and generating a reliability scale on clinical test scores

Savas Okyay[1,2] and Nihat Adar[1]

[1] Computer Engineering, Eskisehir Osmangazi University, Eskisehir, Turkey
[2] Computer Engineering, Eskisehir Technical University, Eskisehir, Turkey

## ABSTRACT

Medical doctors may struggle to diagnose dementia, particularly when clinical test scores are missing or incorrect. In case of any doubts, both morphometrics and demographics are crucial when examining dementia in medicine. This study aims to impute and verify clinical test scores with brain MRI analysis and additional demographics, thereby proposing a decision support system that improves diagnosis and prognosis in an easy-to-understand manner. Therefore, we impute the missing clinical test score values by unsupervised dementia-related user-based collaborative filtering to minimize errors. By analyzing succession rates, we propose a reliability scale that can be utilized for the consistency of existing clinical test scores.
The complete base of 816 ADNI1-screening samples was processed, and a hybrid set of 603 features was handled. Moreover, the detailed parameters in use, such as the best neighborhood and input features were evaluated for further comparative analysis. Overall, certain collaborative filtering configurations outperformed alternative state-of-the-art imputation techniques. The imputation system and reliability scale based on the proposed methodology are promising for supporting the clinical tests.

## INTRODUCTION

Neuropsychiatric disorders, also known as dementia diseases, are among the most critical aging problems. During medical check-ups, brain scans, particularly some clinical test scores, are considered. Hence, corrupted medical images or missing information can cause faulty decisions during the diagnostic process (*Liu et al., 2018*).

The clinical diagnostic process emerges from the patient's physical examination, and if necessary, mini-mental tests are applied. These tests should be organized with the support of a psychologist in a specially prepared environment outside the polyclinic, with sufficient time spared. Otherwise, erroneous results may be obtained. Nevertheless, a specially prepared inspection environment and psychologist support cannot be provided by some medical centers because of the vast number of patients. In this direction, medical doctors

Corresponding author
Savas Okyay, osavas@ogu.edu.tr

have expressed the need for a reliability check of mini-mental test scores using computer-aided methods. If the test scores are questionable, the MRI analysis becomes more critical. Therefore, the decision to treat neuropsychiatric disorders is based on mini-mental tests and visual inspection of the MRI scans.

In neuroimaging science, an unknown diagnosis of dementia is determined mainly through MRI-based neuroanatomical studies. A touchstone book in the literature shows that the dementia work-up is not complete without MRI (*Becker & Giacobini, 1990*). Likewise, markers derived from structural MRI scans can be considered as aids in clinical decision-making and treatment development (*Sabuncu & Konukoglu, 2015*). However, current state-of-the-art MRI machines cannot directly measure a certain region on brain scans in terms of volume, surface area, etc. The output files of scheduled scanning studies can be processed using computer-aided systems by applying image processing techniques or managing software tools. FreeSurfer (*Fischl, 2012*) is one of the most suitable open-source brain analysis tools that takes advantage of MRI. Virtual brain construction from scans and obtaining statistical measurements, that is, morphometric features of particular brain segments, are the most prominent practices. Processing such features with learning algorithms can change or improve the medical horizon to interpret manifestations and paroxysms. Moreover, employing additional demographics may enhance the entire algorithm. Gender, age, and most importantly, the test scores of clinical questionnaires such as mini-mental state examination (MMSE), geriatric depression scale (GDS), clinical dementia rating (CDR), neuropsychiatric inventory questionnaire (NIQ), functional assessment questionnaire (FAQ) can be exemplified within the scope of demographics.

Any sample with incomplete data is a limitation in data science, and preprocessing the data containing missing values is a crossroads for any study. Information may be missing completely at random, missing at random, or missing not at random (*Jabason, Ahmad & Swamy, 2018*). Ignoring the corresponding incomplete samples and filling sparsity have their own advantages and disadvantages. Discarding the samples is the easiest solution. Nevertheless, this option reduces the number of samples (*Zhou et al., 2019a*) and further learning complications may occur. On the other hand, the filling phase can be extremely costly (*Cruz et al., 2020*), and it might not have been accomplished accurately. Basically, in the discarding option, a significant amount of convenient data vanishes intentionally, and in the imputation option, additional noise can be inserted into the information (*Liu et al., 2018*). Consequently, instead of discarding any samples, it is necessary to fill in the missing data in the most appropriate theoretical manner. Thereby, any minor data that are not ignored can be benefited the learning algorithm.

The reliability scale can be defined as the relevance between the clinical test score and computer-aided generated value. An acceptable difference between the two is expected to generate a scale. If the difference is smaller than a threshold, the clinical test score is considered reliable. The focus is on setting this threshold. To the best of our knowledge, no comparable research has generated a proper threshold on computer-aided clinical brain activity test score. However, a confidence interval is based on sampling the distribution of a parameter (*Dekking et al., 2005*). The most widely utilized (*Zar, 1999*) two-sigma

**Table 1 Reliability scale results example.**

| Clinical test type | Range | Clinical test score | Computer-aided prediction | Absolute error percentage[§] | Reliability |
|---|---|---|---|---|---|
| MMSE | [0, 30] | 20 | 22.555 | 8.52 | not trusted |
| MMSE | [0, 30] | 21 | 26.469 | 18.23 | not trusted |
| MMSE | [0, 30] | 28 | 27.977 | 0.08 | trusted |
| GDS | [0, 12] | 0 | 0.662 | 5.52 | trusted |
| GDS | [0, 12] | 2 | 2.01 | 0.08 | trusted |
| GDS | [0, 12] | 5 | 1.335 | 30.54 | not trusted |
| CDR | [−1, 3] | 0.5 | 0.5 | 0 | trusted |
| CDR | [−1, 3] | 0.5 | 0.552 | 1.3 | trusted |
| CDR | [−1, 3] | 1 | 0.166 | 20.85 | not trusted |

**Note:**
[§] Absolute Error Percentage = |Clinical Test Score − Computer-Aided Prediction| / Range

methodology in statistics, which provides a reasonable confidence level of approximately 95% (*Pukelsheim, 1994*; *Coory, Duckett & Sketcher-Baker, 2008*), might yield remarkable results in terms of reliability scaling.

Table 1 presents a reliability scale example with scores randomly taken from the analyses. In addition to the aforementioned actual and predicted values, the ratio of the absolute difference between the clinical test score and computer-aided prediction to the test score range provides the error percentage. The last column, reliability, indicates whether the corresponding clinical test score value is acceptable. If not trusted, all tests should be re-run from the start.

This research has two aims:

The first aim was to impute the missing clinical test scores. Therefore, we focused on providing critical clinical test score values by transforming the MRI measurements and additional information. An unsupervised dementia-related user-based collaborative filtering (DUCF) missing-value imputation methodology with detailed parameter settings was proposed. Because this is an unsupervised algorithm, its objective is to reveal the relationships between morphometrics and demographics to verify the clinical test scores independently of the dementia type. Both feature forms, morphometrics, and demographics, used in clinical diagnostics, were processed individually and as a hybrid combination; then, the enhancement of the results was examined. Similar samples were detected by elaborative test configurations using various features, and the best neighbor count (BNC) was examined in terms of minimizing errors. In the final stage, the imputation values, that is, predictions, were stored as the weighted average of the test scores through similarity. The validation performance of the DUCF configurations as an alternative imputation technique was comparatively measured. The results are promising, and certain DUCF configurations outperformed other state-of-the-art imputation techniques.

The second aim was to generate a reliability scale for clinical test scores. In this context, the measured values of brain sections extracted from MRI scans were transformed into a

chart similar to mini-mental test results, and the margin between these values and clinical test results was calculated. The corresponding predefined threshold with a 95% confidence level was then calculated for each clinical test. We refer to this threshold as the reliability scale. A smaller error on the reliability scale implies more accurate clinical test results. If the difference between the predicted and clinical test score values exceeds the reliability scale, clinical tests should be repeated. Thus, medical doctors can easily interpret processed values based on measurements instead of visually inspecting the MRI scans.

Overall, the following highlights are presented in this research.

- An unsupervised missing value imputation methodology, DUCF with configurations, is proposed to impute missing clinical brain activity test scores.
- There is no comparable brain research in the medical informatics and neuroimaging fields in which collaborative filtering imputation has been inspected in-depth.
- A reliability scale for clinical brain activity test scores is proposed.
- The reliability scale is expected to prevent erroneous future decisions.
- To the best of our knowledge, no comparable research has generated a reliability scale on computer-aided clinical brain activity test score.
- The effect of utilizing additional FreeSurfer morphometrics on collaborative filtering is demonstrated.
- Different normalization strategies were implemented to optimize the data.

The remainder of this study is organized as follows. The "Related work" subsection exemplifying similar studies on state-of-the-art methods is given next. In the "Materials and Methods" section, the dataset containing missing values is first described. Subsequently, the imputation methodology in terms of the missing-value concept and the corresponding configurable parameters are addressed. After analyzing the tests and the outcomes in the "Imputation Tests and Results" section, the conclusion and future plans are indicated in the "Conclusion" section.

## Related work

Considering the medical informatics and neuroimaging fields, incomplete datasets with diversified modalities are processed using methods such as incomplete source-feature selection (iSFS) (*Liu et al., 2018*), incomplete multiview weak-label learning (iMVWL) (*Zhou et al., 2019b*), doubly aligned incomplete multiview clustering (DAIMC) (*Zhou et al., 2019a*, *2019b*), and incomplete multi-source feature (iMSF) (*Liu et al., 2018*; *Zhou et al., 2019a*, *2019b*). Some studies use incomplete multimodality data, and missing data recovery is one of the main purposes (*Zhou et al., 2019b*; *Zhu et al., 2017*; *Thung et al., 2015*; *Thung, Yap & Shen, 2018*). The model can generate the missing modality by taking the existing modality as input (*Cai et al., 2018*), such as any missing neuroimaging data (*Zhou et al., 2019a*, *2020*) or different multimodalities, for example, positron emission tomography (*Liu et al., 2018*). Rather, *Abdelaziz, Wang & Elazab (2021)* fills missing features for each incomplete multimodal sample using convolutional neural networks. In recent research, a deep learning framework combining a task-induced pyramid and

attention generative adversarial network with a pathwise transfer dense convolution network for imputation was proposed, and the missing positron emission tomography data were rendered with their MRI. Utilizing the imputed multimodal images, a dense convolution network was built for disease classification (*Gao et al., 2022*). *Liu et al. (2022)* proposed generative adversarial and classification networks to synthesize missing images, generate multimodal features, and conversion prediction. However, only one modality can have its own sparseness. For instance, structural MRI imputation focus can be through ROIs (*Collazos-Huertas, Cardenas-Pena & Castellanos-Dominguez, 2019*). The lack of pixels/voxels in the brain scans was corrected using FreeSurfer v5.3.0 by *Cruz et al. (2020)*. Block-wise missing data were integrated in *Xue & Qu (2020)*. Feature selection and classification techniques were applied to extract feature matrices from incomplete multimodal high-dimensional data by filtering certain non-sparse data and discarding some column-wise missing values by *Deng, Liu & Dong (2018)*. Considering how accurate it is to shrink the information, that is, wasting the samples, the emphasis of an imputation methodology aiming at increasing efficiency was revealed.

Some straightforward but occasionally effective simple imputation approaches, such as zero value (*Liu et al., 2018*; *Zhou et al., 2019a*, *2020*; *Campos et al., 2015*), attribute mean (*Jabason, Ahmad & Swamy, 2018*; *Cruz et al., 2020*; *Campos et al., 2015*), attribute winsorized mean (*Campos et al., 2015*), and attribute median (*Campos et al., 2015*), stand out to fill the missing values in a dataset. Advanced methods with higher time complexity, require computational performance, such as expectation maximization (EM) (*Liu et al., 2018*), regularised expectation maximisation (RegEM) (*Campos et al., 2015*), low-rank matrix completion (LRMC) or approximation (*Zhou et al., 2019a*; *Cruz et al., 2020*; *Zhou et al., 2019b*; *Thung et al., 2015*), matrix shrinkage and completion (MSC) (*Liu et al., 2018*; *Zhou et al., 2019a*), and multiple imputation using denoising autoencoders (MIDA) (*Jabason, Ahmad & Swamy, 2018*) were also encountered on imputing missing values. Examples of deep learning methods can also be noticed, for example, research on forwarding, linear, and model filling using RNN by *Nguyen et al. (2020)*. Moreover, there are also studies on KNN (*Liu et al., 2018*; *Jabason, Ahmad & Swamy, 2018*; *Zhou et al., 2019a*; *Cruz et al., 2020*; *Zhou et al., 2019b*; *Zhou et al., 2020*; *Campos et al., 2015*), which include distance measurement approaches. However, imputation methods, such as KNN and LRMC, work efficiently when a small portion of the data is missing, and the overall performance declines if the sparsity is high (*Zhou et al., 2019b*).

Considering the missing clinical brain activity test scores and reliability scale on computer-aided clinical test score, there is limited related work to validate actual clinical test scores and/or value imputations. A machine learning approach was proposed to statistically impute cognitive test scores across different datasets for data harmonization. This statistical analysis of normalized scores yields data distributions and allows outcomes from theoretically identical tests (*Shishegar et al., 2021*). Missing clinical test scores at multiple time points were predicted through deep and joint learning by *Lei et al. (2020)*; the absent information was imputed by associating the monthly check values of the corresponding patient with the regression framework. One of the objectives of *Mehta et al. (2022)* is to develop an accurate model to predict Alzheimer's disease clinical scores by

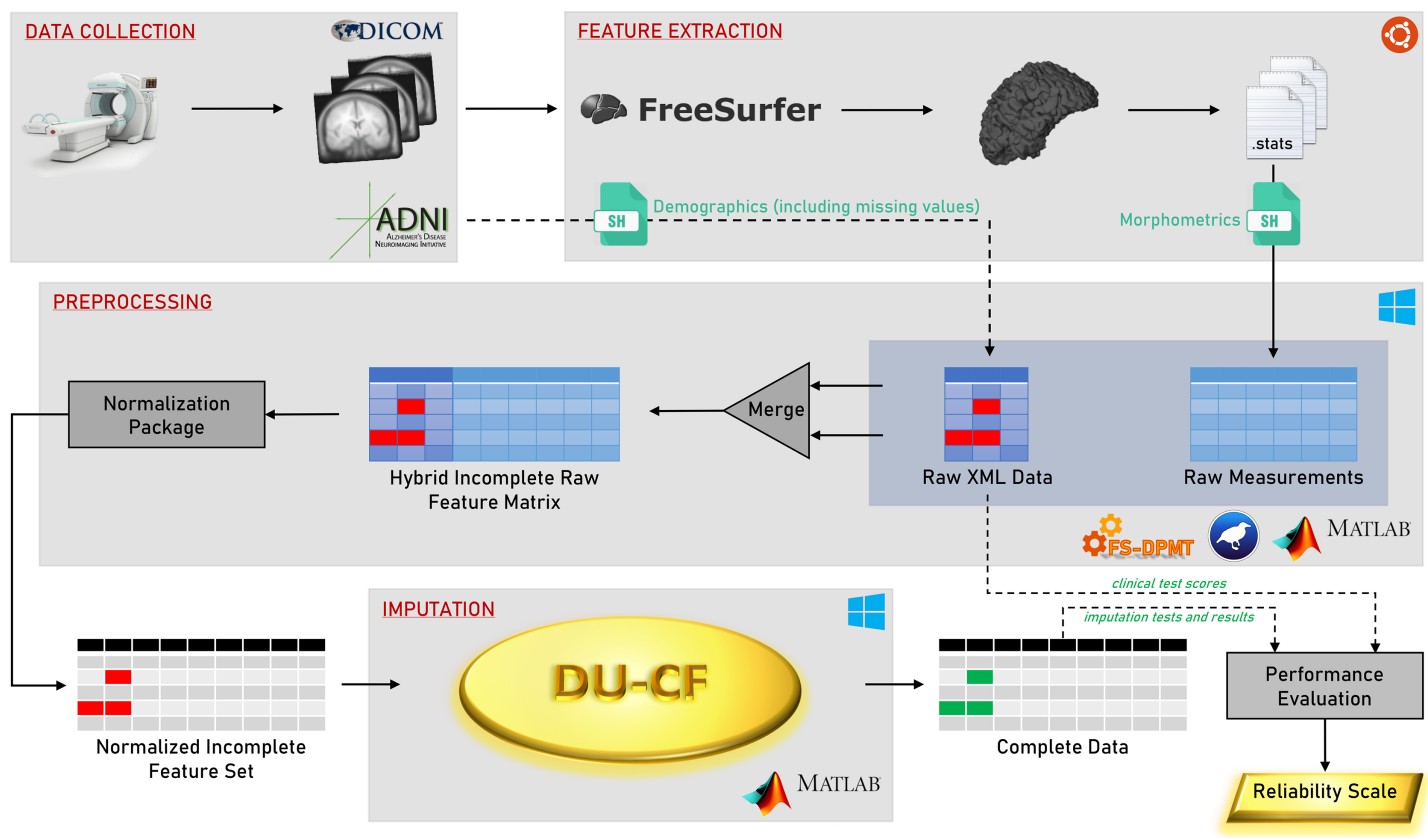

**Figure 1   Graphical flowchart of the methodology.**

embedding uncertainty estimates across cascaded inference tasks; however, the research was conducted in brain tumor segmentation, and there was no imputation discussion. *Abdelaziz, Wang & Elazab (2021)* estimated clinical test scores using supervised convolutional networks as a regression task for various binary diseases and statistically discussed the correlations between the estimated and the actual scores.

Although there are similar studies related to the analysis of frequency of daily living activities with big data using collaborative filtering (*Moldovan et al., 2018*) or computing cohorts' similarity with recommender systems (*Almeida et al., 2020*), there is no comparable brain research in neuroimaging and medical informatics fields, in which recommender systems methodologies or collaborative filtering are inspected in-depth in terms of imputation and/or generating a reliability scale on computer-aided clinical test score.

## MATERIALS AND METHODS

A flowchart of the methodology is shown in Fig. 1. The steps in the graphical representation are detailed in the following subsections.

### Data collection

Data used in the preparation of this article were obtained from the Alzheimer's Disease Neuroimaging Initiative (ADNI) database (adni.loni.usc.edu, ADNI1Complete1Yr1.5T).

**Table 2 Screening schedule and clinical test score sparsity relationship.**

| Screening schedule | Sample count | MMSE | GDS | CDR | NIQ | FAQ |
|---|---|---|---|---|---|---|
| ADNI1 Screening | 828 | ✓ | ✓ | ✓ | | |
| ADNI1/GO Month 6 | 753 | ✓ | | ✓ | ✓ | ✓ |
| ADNI1/GO Month 12 | 710 | ✓ | ✓ | ✓ | ✓ | ✓ |

The ADNI was launched in 2003 as a public-private partnership, led by Principal Investigator Michael W. Weiner, MD. The primary goal of ADNI has been to test whether serial MRI, positron emission tomography, other biological markers, and clinical and neuropsychological assessment can be combined to measure the progression of mild cognitive impairment (MCI) and early Alzheimer's disease (AD). For up-to-date information, see adni-info.org.

The screening scheduling and sparse clinical test score information of the dataset containing 2,291 samples are listed in Table 2. Each row indicates the screening schedule of the brain imaging study, and the columns list the number of corresponding studies and the existence of various clinical test scores. To ignore the impact of continuing screening of any sample during imputation, the analysis of missing critical test score values was evaluated using the complete base *ADNI1 Screening* data. The score ranges of the clinical tests in the dataset, the minimum and maximum values, vary within themselves: [8, 30] for MMSE, [0, 12] for GDS, [−1, 3] for CDR, [0, 29] for NIQ, and [0, 30] for FAQ. Normalized values were used to equalize the scoring scale.

## Feature extraction

Statistical data, that is, morphometrics, from raw brain scans were obtained using FreeSurfer, and a parallel feature extraction study was reported by *Okyay & Adar (2018)*. Except for 816 samples within the base data, the rest did not survive the virtual brain construction process.

The initial feature set included two subparts: morphometrics and demographics. The former consisted of 594 structural MRI attributes after filtering specific FreeSurfer-stats[1] containing keywords, such as general, volumemm3, area, thickavg, grayvol, nvoxels, numvert, and nvertices. Some factors may have a negligible impact on the process. Regardless, all morphometric attributes were included in the initial feature set to boost the interaction between features. The latter provided by ADNI-XMLs was added to the initial feature set after manually eliminating insignificant attributes, such as weight and some referrals. The selected demographics contained nine attributes, namely, sex, age, apoE-A1, and apoE-A2, and the five clinical test scores in Table 2. Therewith, it yielded a total of 603 features[2] for 816 samples. A brief list of the aforementioned initial feature set is presented in Fig. 2.

## Preprocessing

Row- and column-based anomalies in the morphometrics were automatically ignored. The morphometric features in both the right and left parts of the brain were averaged.

[1] Achieving a successful virtual brain construction process creates not only 3D models but also textual-based files comprising the numerical statistics of brain regions.

[2] The complete list of the features, including the details of preprocessing, is attached to the *AllFeaturesList* tab in the supplementary material available at GitHub: https://github.com/savasokyay/Dementia-Related-User-Based-Collaborative-Filtering/blob/main/supplementaryMaterial-DUCF.xlsx.

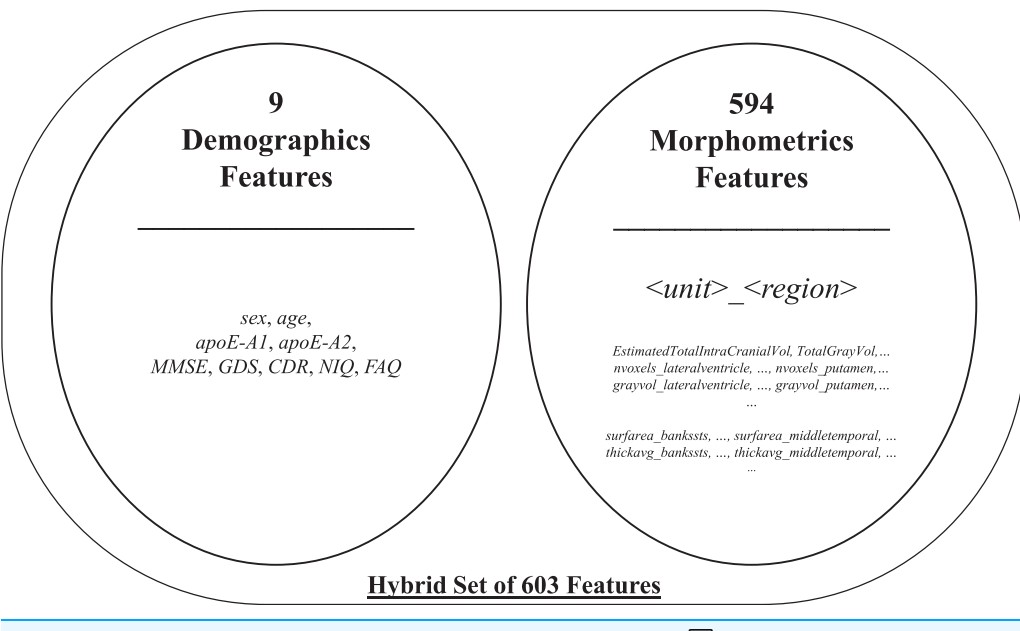

**Figure 2  Brief list of the initial feature set.**     

Additionally, each feature or feature group in the dataset was functionally normalized to the range of [0, 1] using several techniques suitable for the keywords through the normalization package. The volume characteristics were normalized by the skull volume attribute (EstimatedTotalIntraCranialVol), and the gray volumes were normalized by the total gray volume attribute (TotalGrayVol). The min-max normalization method was applied to the remaining attributes, except for those that should not be normalized or require special processing, such as nominal features. During the preprocessing step, FS-DPMT v1.0615 by *Okyay, Adar & Kaya (2020)* and MATLAB R2020b were utilized.

## Imputation
### Other imputation techniques
The proposed methodology was compared with state-of-the-art zero value, attribute mean, attribute winsorized mean, attribute median, RegEM, and LRMC methods mentioned in the introduction section. The KNN method encountered in the literature was included in the tests within the DUCF scope, detailed with the parameter analysis and configuration.

### Dementia-related user-based collaborative filtering
The proposed DUCF imputation design was unsupervised, therefore it was not affected by class differences, thus, conducted a pure analysis by interacting attributes in the most exposed manner. The methodology consisted of two stages: calculating user-based similarities and predicting missing values.

User-based similarity weights, $w$, were obtained using five equations commonly used in recommender system terminology and collaborative filtering methodology. These are correlation-based (*Choi, Cha & Tappert, 2010*) Pearson Correlation Coefficient (PCC)

**Table 3 Similarity weight and prediction equations in practice.**

| Stage (type) | Name | Equation | |
|---|---|---|---|
| Similarity (corr.) | Pearson Correlation Coefficient | $$w_{x,y}^{PCC} = \frac{\sum_{i=1}^{n}(x_i - \bar{x})(y_i - \bar{y})}{\sqrt{\sum_{i=1}^{n}(x_i - \bar{x})^2}\sqrt{\sum_{i=1}^{n}(y_i - \bar{y})^2}}$$ | (3.1) |
| Similarity (corr.) | Median-Based Robust Correlation | $$w_{x,y}^{MRC} = \frac{\sum_{i=1}^{n}(x_i - \tilde{x})(y_i - \tilde{y})}{\sqrt{\sum_{i=1}^{n}(x_i - \tilde{x})^2}\sqrt{\sum_{i=1}^{n}(y_i - \tilde{y})^2}}$$ | (3.2) |
| Similarity (corr.) | Cosine Similarity | $$w_{x,y}^{COS} = \frac{\sum_{i=1}^{n} x_i y_i}{\sqrt{\sum_{i=1}^{n}(x_i)^2}\sqrt{\sum_{i=1}^{n}(y_i)^2}}$$ | (3.3) |
| Similarity (dist.) | Manhattan Distance Similarity | $$w_{x,y}^{MAN} = 1/\left(\sum_{i=1}^{n}|x_i - y_i|\right)$$ | (3.4) |
| Similarity (dist.) | Euclidian Distance Similarity | $$w_{x,y}^{EUC} = 1/\left(\sqrt{\sum_{i=1}^{n}(x_i - y_i)^2}\right)$$ | (3.5) |
| Prediction (avg.) | Weighted Average | $$p_{x,cs} = \frac{\sum_{y^{\triangleright}=1}^{BNC} y_{cs}^{\triangleright} \times w_{x,y^{\triangleright}}^{*}}{\sum_{y^{\triangleright}=1}^{BNC} w_{x,y^{\triangleright}}^{*}}$$ | (3.6) |

(*Polatidis & Georgiadis, 2016*; *Shi, Larson & Hanjalic, 2014*) in Eq. (3.1), Median-based Robust Correlation coefficient (MRC) (*Shevlyakov, 1997*; *Shevlyakov & Smirnov, 2011*) in Eq. (3.2), COSine similarity (*Adomavicius & Tuzhilin, 2005*) in Eq. (3.3), and distance-based (*Choi, Cha & Tappert, 2010*) MANhattan distance similarity (MAN) in Eq. (3.4), and EUClidian distance similarity (EUC) in Eq. (3.5). The details are summarized in Table 3, where $x$ and $y$ are the sample vectors, and $\bar{x}$ and $\tilde{x}$ are the corresponding mean and median of $x$, respectively. The maximum correlation is denoted by +1, while negative ones end up through −1 in the range of [−1, +1]. Likewise, because distance is a negating factor, the computed inverse proportional similarity weight became a positive value. Although rare conditions exist when the same vector values coincide in the similarity calculation, the distance-based weight was infinite. In this case, the highest similarity computed as a real number was assigned to the corresponding value.

In the prediction stage, patients were filtered to the most similar neighbors with higher similarity weights. Then, the prediction, $p$, for any missing test score value was computed over the weighted average formulation, as in Eq. (3.6), where $w^{*}$ is the weights for one of the similarity equations, $x$ is the patient-of-interest, $cs$ represents the clinical test score type, $y^{\triangleright}$ contains vectors sorted by one of the similarity equations, and $BNC$ describes the top-$N$ similar patients that took place in the computation.

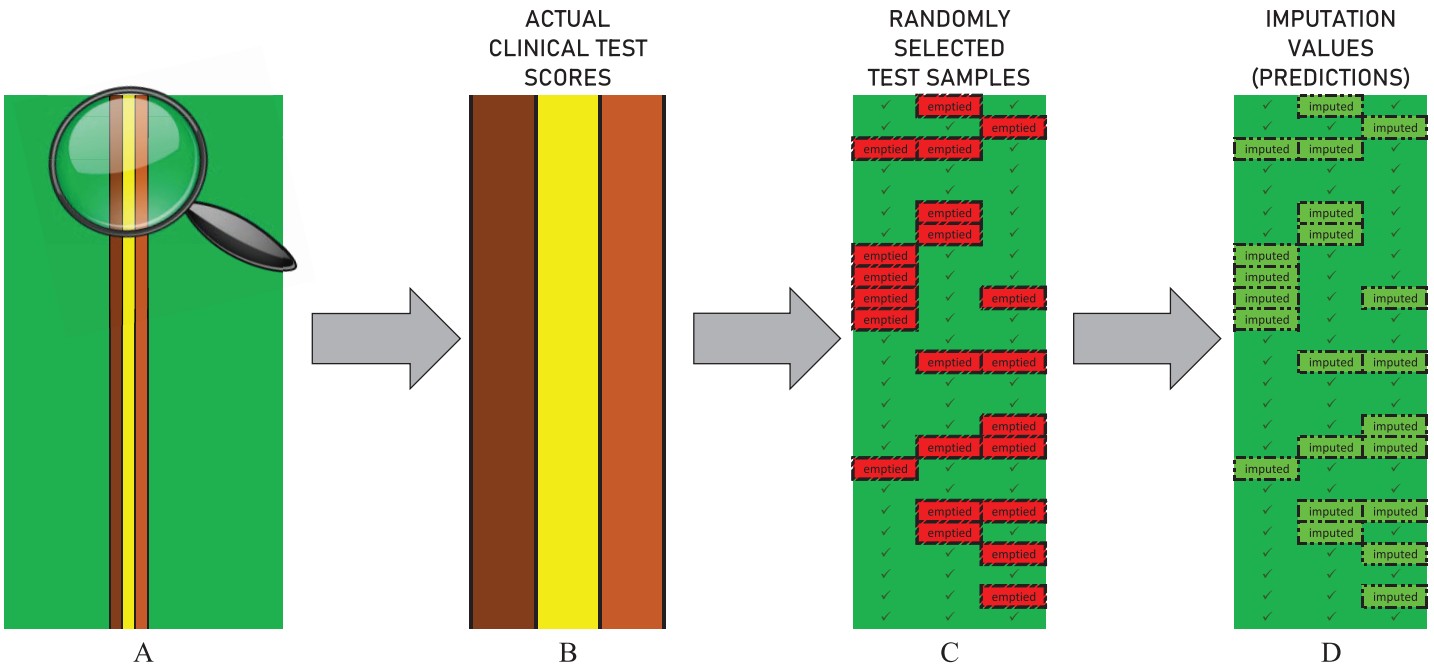

**Figure 3  Illustration of the data elimination and imputation operations in the test procedure.** The figure represents the actual values of the first 25 samples in a randomly selected independent test scenario. (A) The non-green vertical lines indicate the columns of the non-sparse clinical test score attributes. (B) The zoomed-in representation of corresponding vertical column values is shown. (C) The bordered red cells indicate that the related values are randomly emptied. (D) The bordered light green cells indicate that the missing values are imputed.

## IMPUTATION TESTS AND RESULTS

The performance of all imputation techniques covered in the test procedure, including DUCF, was evaluated by eliminating partial data. Each clinical test score attribute in demographics was emptied for 250 randomly selected samples[3] as illustrated in Fig. 3. Hence, an independent test procedure is based on a stochastic behavior, that is, the principle of working with random values. Therefore, the procedure was repeated 100 times to optimize the randomness effect and provide stable outcomes. The average of all individual test results is reported for all approaches.

For each missing clinical test score value when applying DUCF, leave-one-out cross-validation was performed. The similarities between the sample of interest and all other instances were found in the first stage through all independent test combinations. A DUCF test combination consisted of five unique similarity equations, varying BNCs (from one to 100 incremented by one), and three types of feature vectors. Therefore, morphometrics and demographics within the initial feature set were included in the tests in the following three forms.

1. Only morphometrics: 594 features from FreeSurfer stats.

2. Only demographics: nine features from the ADNI XMLs.

3. Hybrid set: a total of 603 features.

[3] A complete imputation example showing the sparse matrices for both selected test samples and imputed values is provided in the *ImputationExample* tab of the supplementary material available at GitHub: https://github.com/savasokyay/Dementia-Related-User-Based-Collaborative-Filtering/blob/main/supplementaryMaterial-DUCF.xlsx.

# PeerJ

Actual values define the ground truth, regardless of the initial feature set and methodology. Imputed values, that is, predictions, were obtained by applying imputation techniques. The two matrices, the actual and imputed values, are of the same size and have identical sparsity[3], in which the same indices are filled.

In this study, actual clinical test scores and estimated imputation values were statistically compared. For this purpose, the correlation results for all individual tests were computed, analyzed, and given in the supplementary material[4]. Moreover, paired-samples t-tests were deliberately performed for the selected important test configurations, where the trusted and moderate thresholds for the reliability scale were mainly determined. Full statistical reports are also included[4] in the supplementary material.

It is mentioned in the study that imputation tests were executed many times for the same configuration. The weakest imputation set of the relevant test configuration, most outlying the mean, was selected as a case example. This is because other executions would theoretically produce adequate results. The statistical results for two[5] of the selected important configurations are reported below.

1. Results of the correlation analysis indicated that there was a significant positive association between clinical test scores and DUCF imputation ($r(750) = 0.934$, $p < 0.000$). A paired-samples t-test was conducted, and there was no significant difference between actual clinical test scores ($M = 0.44$, $SD = 0.327$) and estimated imputation values ($M = 0.444$, $SD = 0.34$); $t(749) = -0.904$, $p = 0.366$.

2. Results of the correlation analysis indicated that there was a significant positive association between clinical test scores and RegEM imputation ($r(750) = 0.960$, $p < 0.000$). A paired-samples $t$-test was conducted, and there was no significant difference between actual clinical test scores ($M = 0.448$, $SD = 0.329$) and estimated imputation values ($M = 0.443$, $SD = 0.316$); $t(749) = 1.529$, $p = 0.127$.

After statistical analyses between the actual and predicted values, all tested missing value imputation approaches were evaluated using these two value sets over error and regression metrics as mean absolute error (MAE) in Eq. (4.1), mean squared error (MSE) in Eq. (4.2), root mean squared error (RMSE) in Eq. (4.3), and r-squared ($R^2$) in Eq. (4.4). Lower values are important for the error metrics. A higher value of $R^2$ indicates a better prediction accuracy. The performance metric formulations are listed in Table 4, where $x$ and $y$ are the actual and imputed values, respectively.

The internal configurations and optimal parameters of the DUCF imputation methodology are shown in Fig. 4. The grid-type figure contains 12 subplots ($3 \times 4$: three rows and four columns) containing various test and evaluation parameters. Each row of the subplots matches the initial feature set enumeration. Each column was settled based on the performance metrics in Eqs. (4.1)–(4.4). For all subplots, the $x$-axis defines the corresponding BNC in each independent test, while the $y$-axis represents the performance metric outcome. The most applicable DUCF technique was inferred by interpreting the configurations.

[4] Correlation details and full statistical reports for paired-samples *t*-tests can be found in the *correlations* and *fullStatisticalReports* tabs of the supplementary material available at GitHub: https://github.com/savasokyay/Dementia-Related-User-Based-Collaborative-Filtering/blob/main/supplementaryMaterial-DUCF.xlsx.

[5] The supplementary material available at GitHub: https://github.com/savasokyay/Dementia-Related-User-Based-Collaborative-Filtering/blob/main/supplementaryMaterial-DUCF.xlsx stores these outcomes in the *fullStatisticalReports* tab, labeled as (1) *DUCF_2W* and (2) *RegEM_4W*.

**Table 4 Performance metric formulations in practice.**

| Name | Formula | |
|---|---|---|
| Mean Absolute Error | $mae(x, y) = \left(\dfrac{1}{n}\right) \sum_{i=1}^{n} \lvert y_i - x_i \rvert$ | (4.1) |
| Mean Squared Error | $mse(x, y) = \left(\dfrac{1}{n}\right) \sum_{i=1}^{n} (y_i - x_i)^2$ | (4.2) |
| Root Mean Squared Error | $rmse(x, y) = \sqrt{\left(\dfrac{1}{n}\right) \sum_{i=1}^{n} (y_i - x_i)^2}$ | (4.3) |
| R-Squared | $rsq(x, y) = 1 - \dfrac{\sum_{i=1}^{n} (y_i - x_i)^2}{\sum_{i=1}^{n} (\bar{x} - x_i)^2}$ | (4.4) |

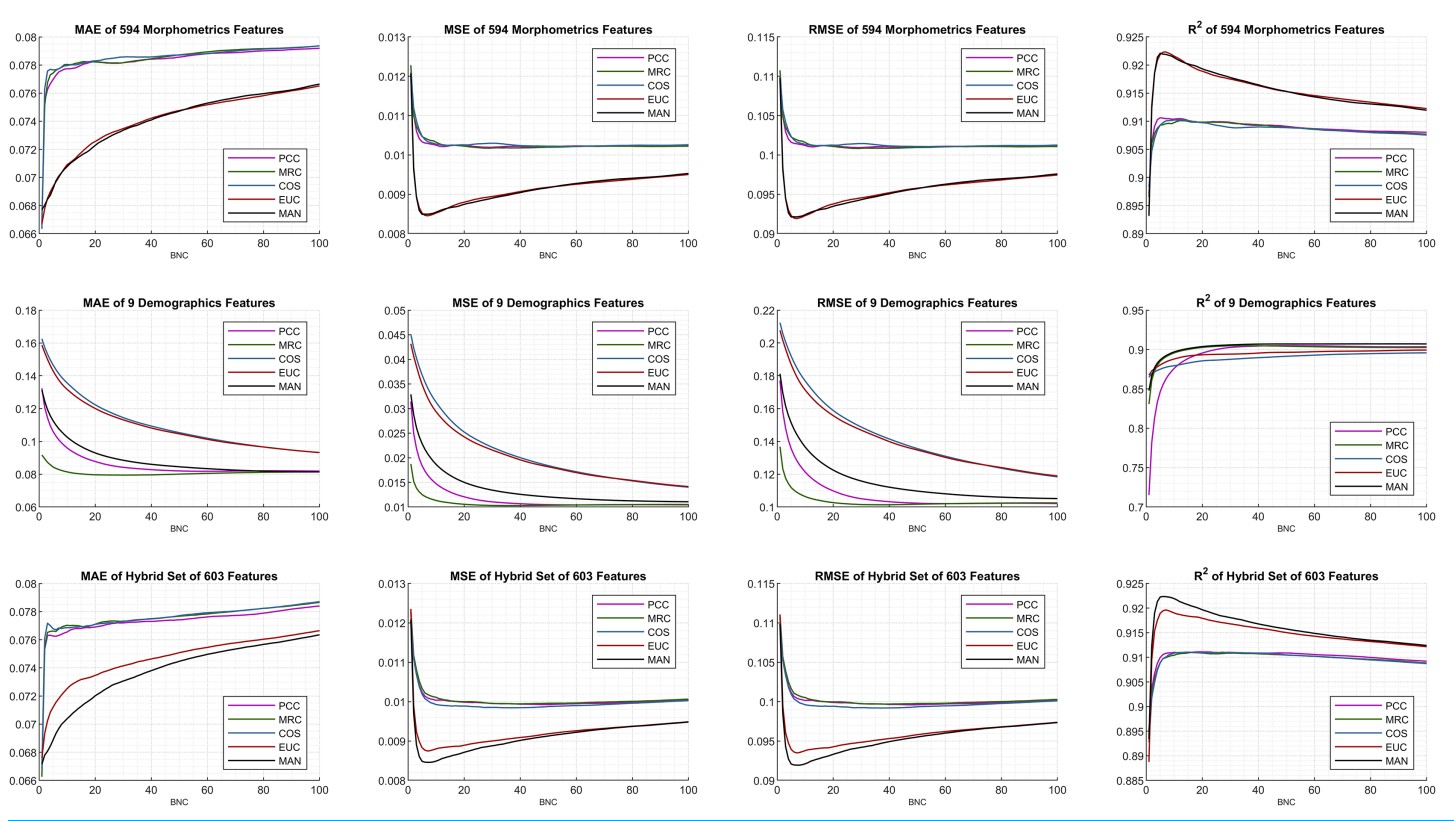

**Figure 4 Performance metric plots of the average of multiple individual tests.**

Considering the similarity equation-type perspective, the plot lines seem to be commonly clustered, in which the black and red lines are separated from the others. Although correlation-based similarities produce similar effects, distance-based similarities perform precisely well for all performance metrics when efficient features are selected.

The initial features were examined, and morphometrics alone performed better than the choice of only demographics for all performance metrics. This implication leads to another explanation that only demographics (subplots in the middle row) are not sufficient. Basically, a few features may not be sufficient to minimize errors. Even though the feature size increased, the interaction between the features also increased, bringing the hybrid set evaluation to the most efficient point. As a mark, the proposed algorithm was not affected by dimensionality and continues to function thoroughly. Hence, the effect of utilizing additional and supplementary morphometrics extracted using FreeSurfer on DUCF is explicitly demonstrated.

On the other hand, including morphometrics in the initial features worsened the results as the BNC increases for the MAE performance metric. Moreover, the error rate of the correlation-based equations increased abruptly after a certain small number of BNC. The best MAE results were achieved with less BNC and the hybrid set of features. In terms of the MSE and RMSE, the best outcome appeared when using the hybrid set of features, particularly for distance-based similarities. The 1-norm distance, MAN, strictly outperformed the others, because it is observed that the black-colored lines reach the most efficient points when employing the corresponding performance metrics with the hybrid set of features. The enhancing trend for both distance-based similarities appeared to improve a certain BNC threshold. After this point, the slope reverses.

According to these performance metrics and configurations, the algorithms reached the most effective points for approximately less than ten BNCs. The exact BNCs are discussed later in the table of results. The behavior of the last performance metric, $R^2$, owing to the BNC was similar to the squared error metrics MSE and RMSE. As diverse behaviors are observed in the performance of related metrics, it can be concluded that performance metrics should be examined together instead of focusing on only one of them.

BNC-oriented DUCF performance results and reference results obtained using other missing data imputation methods in the literature were inspected accordingly. Selected best-performing test results[6] were filtered in the color-mapped Table 5. Each row of the table represents an imputation method with a specific configuration and corresponding mean results of the same iteration in the repeated test package. The overall sorted results are the summation-based combination of the performance metric rankings owing to the instability of the column shades of any metric. For instance, DUCF approaches with low BNCs performed the best in terms of MAE. Another example is the case for RegEM and LRMC, which have more green shading for RMSE and $R^2$. In the plot interpretations, it was stated that the enhancing trend might stop after a certain BNC threshold. However, in the table of results, this interpretation can be inferred as a deviation in the user data.

Based on the colorized results in the table, each performance metric provided a prediction bias. Therefore, the most convenient range for the reliability scale, in accordance with the generated model, should be considered as the greenest shade in each column. Variable thresholds were determined through two-sigma of all performance metrics. Thus, we specified the reliability scale in two parts: trusted and moderate. Utilizing

[6] The complete detailed results of the whole tests can be found in the *DetailedImputationsResults* tab of the supplementary material available at GitHub: https://github.com/savasokyay/Dementia-Related-User-Based-Collaborative-Filtering/blob/main/supplementaryMaterial-DUCF.xlsx.

**Table 5 Selected best-performing test results ordered by the summation-based combination of the performance metric rankings.**

| Imputation technique | Feature vector set | Similarity measurement | BNC | MAE | RMSE | R² |
|---|---|---|---|---|---|---|
| DUCF | Hybrid Set of Features | MAN | 6 | 0.06933 | 0.09196 | 0.92235 |
| DUCF | Hybrid Set of Features | MAN | 7 | 0.06968 | 0.09194 | 0.92232 |
| DUCF | Hybrid Set of Features | MAN | 5 | 0.06888 | 0.09208 | 0.92225 |
| RegEM | Hybrid Set of Features | - | - | 0.06988 | 0.08956 | 0.92546 |
| DUCF | Morphometrics Features | EUC | 7 | 0.07000 | 0.09192 | 0.92232 |
| DUCF | Morphometrics Features | EUC | 6 | 0.06968 | 0.09197 | 0.92228 |
| DUCF | Morphometrics Features | MAN | 5 | 0.06921 | 0.09211 | 0.92210 |
| DUCF | Hybrid Set of Features | MAN | 8 | 0.06998 | 0.09194 | 0.92225 |
| LRMC | Hybrid Set of Features | - | - | 0.07109 | 0.09216 | 0.92155 |
| DUCF | Hybrid Set of Features | EUC | 6 | 0.07155 | 0.09360 | 0.91945 |
| DUCF | Morphometrics Features | EUC | 2 | 0.06778 | 0.09808 | 0.91280 |
| DUCF | Hybrid Set of Features | MAN | 2 | 0.06780 | 0.09851 | 0.91236 |
| DUCF | Morphometrics Features | EUC | 1 | 0.06670 | 0.10956 | 0.89329 |
| DUCF | Morphometrics Features | COS | 1 | 0.06634 | 0.11000 | 0.89831 |
| DUCF | Hybrid Set of Features | MAN | 1 | 0.06714 | 0.10987 | 0.89339 |
| DUCF | Hybrid Set of Features | PCC | 1 | 0.06633 | 0.11064 | 0.89514 |
| DUCF | Hybrid Set of Features | MRC | 1 | 0.06626 | 0.11107 | 0.89418 |
| Attribute Median | Missing Features | - | - | 0.07683 | 0.10673 | 0.89698 |
| Attribute Mean | Missing Features | - | - | 0.08305 | 0.10335 | 0.90032 |
| Attribute Winsorized Mean | Missing Features | - | - | 0.08325 | 0.10423 | 0.89998 |
| Zero Value | Missing Features | - | - | 0.44131 | 0.54957 | - |

**Note:**
Existing methods are highlighted with a gray background color.

**Table 6 Predefined thresholds for the reliability scale.**

| Clinical test type | ±Trusted threshold | ± Moderate threshold |
|---|---|---|
| MMSE | 2.162 | 2.945 |
| GDS | 1.179 | 1.606 |
| CDR | 0.393 | 0.535 |
| NIQ | 2.850 | 3.882 |
| FAQ | 2.948 | 4.016 |

denormalized error rate values, these thresholds can be determined, as summarized in Table 6.

Considering all of these highlights, the outcomes of the DUCF configurations are strictly better than those of the simple imputation approaches, as expected, and are quite analogous to the state-of-the-art methods. Further, DUCF outperformed the other imputation techniques when assessing the MAE and the combination of the performance metric rankings. Additionally, DUCF with suitable configurations, first and foremost, the BNC parameter in a similarity equation, was preferable to other techniques owing to its

efficient computation capability. Consequently, the proposed methodology can be used to reduce possible error rates and improve the reliability scale.

## CONCLUSION

Neuroimaging studies and clinical diagnosis may strain when some morphometrics or demographics, particularly clinical test score attributes, are missing or incorrect. In this study, we imputed the missing clinical test score values by unsupervised dementia-related user-based collaborative filtering to facilitate the studies of neuroimaging and medical informatics researchers, particularly medical doctors. Validation and analysis of the differences between clinical test scores and automatically generated test scores were used as a scale for the confidence of clinical tests. We proposed a reliability scale for computer-aided clinical brain activity test scoring. This may prevent future errors in prediagnoses based on clinical tests and/or visually inspected MRI scans. Furthermore, the effect of utilizing FreeSurfer morphometrics over neuroimaging studies was strengthened by questioning various feature inputs. The input vectors were optimized using several normalization strategies. The detailed configurations of DUCF were evaluated and compared to state-of-the-art imputation techniques in the literature, and its performance was demonstrated. Certain collaborative filtering configurations outperformed other imputation techniques. Then, proper BNCs that could be parameterized directly for further analyses were procured.

The outcomes of the proposed imputation methodology led to some points and are generally promising. This is preferable, particularly because of its computational performance and dimensionality robustness. Consequently, a decision support system with a reliability scale was proposed to impute and verify clinical test scores. Future studies should compare the outcomes of this study with those of incomplete data performance. Dementia-related collaborative filtering with deep-learning techniques may also be considered as a draft plan.

## ACKNOWLEDGEMENTS

We thank neurologist Dr. Serdar Eren (LÖSANTE) for providing us with information about the clinical procedures and explaining the need for computer-aided reliability.

### Funding

This research was not funded by any initiative, including the Alzheimer's Disease Neuroimaging Initiative.

### Competing Interests

The authors declare that they have no competing interests.

## Author Contributions

- Savas Okyay conceived and designed the experiments, performed the experiments, analyzed the data, prepared figures and/or tables, authored or reviewed drafts of the paper, performed the computation work, and approved the final draft.
- Nihat Adar conceived and designed the experiments, analyzed the data, authored or reviewed drafts of the paper, and approved the final draft.

## Human Ethics

The following information was supplied relating to ethical approvals (*i.e.*, approving body and any reference numbers):

Alzheimer's Disease Neuroimaging Initiative: "All ADNI data are shared without embargo through the LONI Image and Data Archive (IDA), a secure research data repository. Interested scientists may obtain access to ADNI imaging, clinical, genomic, and biomarker data for the purposes of scientific investigation, teaching, or planning clinical research studies. Access is contingent on adherence to the ADNI Data Use Agreement and the publications' policies outlined in the documents listed below. Note: documents are subject to updates by ADNI." More information is available at http://adni.loni.usc.edu/data-samples/access-data/.

## Data Availability

The project repository containing the supplementary material and other sources are available at GitHub: https://github.com/savasokyay/Dementia-Related-User-Based-Collaborative-Filtering. The supplementaryMaterial-DUCF.xlsx file contains detailed technical information and results.

The data is available at http://adni.loni.usc.edu/data-samples/access-data/.

Data collection and sharing for this project was provided by the Alzheimer's Disease Neuroimaging Initiative (ADNI) (National Institutes of Health Grant U01 AG024904) and DOD ADNI (Department of Defense award number W81XWH-12-2-0012). ADNI is funded by the National Institute on Aging, the National Institute of Biomedical Imaging and Bioengineering, and through generous contributions from the following: AbbVie, Alzheimer's Association; Alzheimer's Drug Discovery Foundation; Araclon Biotech; BioClinica, Inc.; Biogen; Bristol-Myers Squibb Company; CereSpir, Inc.; Cogstate; Eisai Inc.; Elan Pharmaceuticals, Inc.; Eli Lilly and Company; EuroImmun; F. Hoffmann-La Roche Ltd and its affiliated company Genentech, Inc.; Fujirebio; GE Healthcare; IXICO Ltd.; Janssen Alzheimer Immunotherapy Research & Development, LLC.; Johnson & Johnson Pharmaceutical Research & Development LLC.; Lumosity; Lundbeck; Merck & Co., Inc.; Meso Scale Diagnostics, LLC.; NeuroRx Research; Neurotrack Technologies; Novartis Pharmaceuticals Corporation; Pfizer Inc.; Piramal Imaging; Servier; Takeda Pharmaceutical Company; and Transition Therapeutics. The Canadian Institutes of Health Research is providing funds to support ADNI clinical sites in Canada. Private sector contributions are facilitated by the Foundation for the National Institutes of Health (www.fnih.org). The grantee organization is the Northern California Institute for Research

and Education, and the study is coordinated by the Alzheimer's Therapeutic Research Institute at the University of Southern California. ADNI data are disseminated by the Laboratory for Neuro Imaging at the University of Southern California.

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
