# Peer review of "Dementia-related user-based collaborative filtering for imputing missing data and generating a reliability scale on clinical test scores"

_PeerJ, doi:10.7717/peerj.13425_

## Round 0.1 · original submission · Major Revisions

As the authors will realize, the reviewers have major concerns regarding the methodology and findings of the study. The content and flow of the manuscript also need to be substantially revised to increase its readability. Specifically, the language used in the manuscript must be improved therefore I recommend the authors consider professional English editing services.

Reviewer 1 ·

Basic reporting

The author studied generating a reliability scale for clinical brain activity test scores based on medical scans and demographic data. The idea of proposing a decision support system is not new as some papers have already been published by using different methods; the only novelty is the use of an unsupervised missing value imputation methodology. The presentation is not organized logically. Also, the quality of written English is poor, I would suggest the authors have their manuscripts checked by a proficient English language speaker.

Some suggestions for further improvement:

The summary abstract is not concise. I do recommend the re-writing of the abstract and grammar check through the document.

The Introduction section is well-written, and I propose to schematize all the discussion in several paragraphs (to facilitate the reading): (1) motivations, (2) the overall approach, (3) main contributions.
However, the main challenges to the field and the needs and benefits of this study are missed in the introduction.

Many important and recent papers are ignored by the authors. This research area is not new and there are many other papers in the literature.

A "Related Work" section is completely missing (much could be extracted from the introduction), which highlights the most important techniques (at least for the chosen problem), and above all a comparison with the technique proposed in the paper. A final table summarizing the technical aspects of the various methods would be very useful.

A general discussion of the limitations and expectations should be inserted.

Experimental design

The 'unsupervised' approach used in the comparisons is not explained or discussed. There is a brief mention that it can be intuitively deduced in a referenced work, but at least a basic description should also be present in this work. Also, and more importantly, there is no state-of-the-art for generating a reliability scale test scores method considered in comparisons. If there is not something specific to clinical brain activity data, I believe there are several general domains with good performance that could be considered.

The authors need to compare their results with other existing methods and add a table of comparison to the article. In addition, a statistical test should be made.

Validity of the findings

The authors didn't have performed a sufficient analysis of the uncertainties in their results, for example, the numbers of significant figures and tables are not justified. The need for statistical analysis is required to make statements of the significance of differences in results. The authors have to explicitly study the uncertainties of the experimental variables that represent the key to the measurements. The performance metric plots results are not clear. Please explain Figure 4 in the text.

Additional comments

Line 221 - 240: The paragraph is really too long. It makes the meaning difficult to understand. It can be divided at least into 3 sentences.

Reviewer 2 ·

Basic reporting

The article validated an efficient method of the imputation of missing clinical test score values i.e. unsupervised DUCF method. This can enhance the application of machine learning in the field of medical science. The article is well structured with clear and unambiguous professional English. I have some minor queries.
1. Provide the equation to calculate the Absolute error percentage. The information provided in the manuscript is a bit confusing.

Experimental design

1. As DUCF consists of two stages, one is calculating user-based similarities and the other is predicting the missing values, Although the authors have mentioned the approaches for the first step in detail but did not provide enough information regarding the methodology used to predict the missing values based on the similarities.

Validity of the findings

1. As mentioned in line 165, the number of independent parameters is 603 while the number of samples is 816. As the number of parameters is very high so it can cause a Curse of dimensionality, any comments on this.
2. In line 235, the authors have reported that 1-norm distance i.e. MAN performs better. Provide the explanation behind this observation.

Reviewer 3 ·

Basic reporting

This paper proposed a new method to impute the missing clinical test scores using unsupervised dementia-related user-based collaborative filtering, and validate the reliability of predicted scores using MRI analysis. The study itself is innovative, and is within the interest of medical research field. However, the aim of this study should be justified at first, that is, why the MRI analysis can be used to validate the predicted clinical scores. Some details about the prediction method should be specified.

Experimental design

1. There are two aims in this paper, 1) impute clinical test scores, and 2) validate the reliability of imputed scores using MRI and demographics analysis. However, the title “Generating a reliability scale for clinical brain activity test scores based on medical scans and demographic data” only mentions the first aim, which is actually described less than the second aim. I suggest the author to change the title by including both aims.
2. Line 198, Page 6. It is unclear to me how the missing score is predicted. Please add more details about this.
3. Line 135, Page3. “…, the ratio of the absolute difference between two to the test score range gives the error percentage.” It is unclear that which ratio was used, ratio between absolute difference and “two to the test score range”, or the ratio between two and the test score range? What does “two” mean in this sentence?

Validity of the findings

A fundamental question of this study is the rationale of verifying the imputed/predicted missing clinical test scores with morphometrics and demographics. The author should firstly justify why the MRI analysis can be used to evaluate the reliability of the imputed clinical scores. In other words, how are the MRI analysis metrics correlated with full clinical test scores (without removing data)?

Additional comments

1. There are too many abbreviations throughout the paper, which are not commonly used. Please consider to eliminate the unnecessary abbreviations to make the paper easy to read. For example, “recommender systems” (RS), collaborative filtering (CF).
2. Line 165, Page 5. “it is yielded” should be “it yields”.
3. Line 174, Page 5. “is required” should be “requires”.

---

## Round 0.2 · accepted · Accept

The authors successfully addressed all reviewers' comments, and the manuscript is now acceptable in its current form. I congratulate the authors for their work.

Reviewer 1 ·

Basic reporting

The authors addressed all the comments. Therefore, the manuscript is acceptable in its present form.

Experimental design

The authors addressed all the comments. Therefore, the manuscript is acceptable in its present form.

Validity of the findings

The authors addressed all the comments. Therefore, the manuscript is acceptable in its present form.

Additional comments

The authors addressed all the comments. Therefore, the manuscript is acceptable in its present form.

Reviewer 2 ·

Basic reporting

All the mentioned changes have been incorporated into the revised manuscript. It presents clean and professional English throughout with sufficient field background literature survey.

Experimental design

The research is within the Aims and scope of the journal. The manuscript has provided a new methodology i.e., Dementia-related user-based collaborative filtering for filling the missing values. This work would have a high impact to improve the accuracy of the models.

Validity of the findings

The underlying data provided is statistically sound and robust. The conclusions are well stated.

Reviewer 3 ·

Basic reporting

I thank the reviewer for addressing my comments. I am satisfied and have no further comments.

Experimental design

No comments.

Validity of the findings

No comments.

Additional comments

No comments.